# OrcaLoca: An LLM Agent Framework for Software Issue Localization

**Zhongming Yu** [* 1]  **Hejia Zhang** [* 1]  **Yujie Zhao** [1]  **Hanxian Huang** [1]  **Matrix Yao** [2]  **Ke Ding** [2]  **Jishen Zhao** [1]

## Abstract

Recent developments in Large Language Model (LLM) agents are revolutionizing Autonomous Software Engineering (ASE), enabling automated coding, problem fixes, and feature improvements. However, localization – precisely identifying software problems by navigating to relevant code sections – remains a significant challenge. Current approaches often yield suboptimal results due to a lack of effective integration between LLM agents and precise code search mechanisms. This paper introduces ORCALOCA , an LLM agent framework that improves accuracy for software issue localization by integrating priority-based scheduling for LLM-guided action, action decomposition with relevance scoring, and distance-aware context pruning. Experimental results demonstrate that ORCALOCA becomes the new open-source state-of-the-art (SOTA) in function match rate (65.33%) on SWE-bench Lite. It also improves the final resolved rate of an open-source framework by 6.33 percentage points through its patch generation integration. ORCALOCA is available at https://github.com/fishmingyu/OrcaLoca.

## 1. Introduction

Large Language Models (LLMs) have advanced rapidly, driving intelligent agents across diverse domains. In Autonomous Software Engineering (ASE) (Devin, 2024), LLM-driven agents enable automatic code generation, program repair, and feature enhancement. Incorporating LLMs into software development processes has been demonstrated promising by tools such as GitHub Copilot (Microsoft, 2023) and LLM-based agents like AutoCodeRover (Zhang et al., 2024b) and SWE-agent (Yang et al., 2024b). To navigate repositories, create patches, and fix problems, these agents

---
[*]Equal contribution  [1]University of California, San Diego, USA  [2]Intel Corporation. Correspondence to: Jishen Zhao <jzhao@ucsd.edu>.

*Proceedings of the 42$^{nd}$ International Conference on Machine Learning*, Vancouver, Canada. PMLR 267, 2025. Copyright 2025 by the author(s).

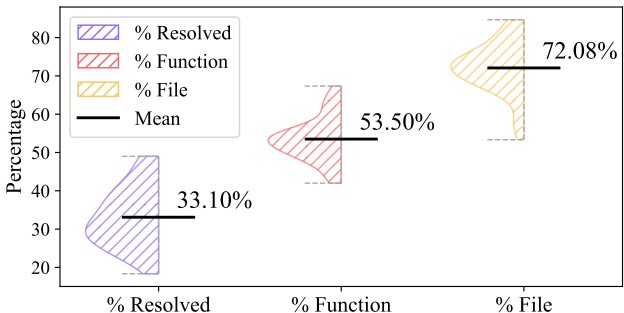

*Figure 1.* Distribution and average of file / function match rate and resolved rate on SWE-Bench Lite LeaderBoard.

leverage capabilities such as fault localization, action planning, and program-building unit tests. Among these abilities, localization – the ability to precisely identify and navigate to relevant code for resolving software engineering problems – remains a crucial yet underexplored challenge in ASE.

Localization is well-recognized as a critical yet challenging step (Yang et al., 2024b; Xia et al., 2024) in ASE. As shown in Figure 1, on average, only 53.5% of issues achieve a correct function match across all submitted agents solutions (Jimenez et al., 2025). Localization is challenging due to an inherent complexity of software repositories. For instance, the average codebase of SWE-bench (Jimenez et al., 2024) consists of 3,010 files with around 438K lines of code.Worse yet, user requirements are often expressed in imprecise natural language, making it even more challenging to extract relevant code from a large repository based on the user's issue input. In particular, we identify three key challenges of LLM agent-based localization:

1) *How to explore the codebase with strategic action planning and precise navigation?* Prior works on agent-based software localization encounter two key limitations: (i) action planning inefficiencies arise as certain methods rely solely on LLMs for guidance (Zhang et al., 2024a), resulting in unstable and redundant search behaviors; (ii) graph-based scheduling (Ma et al., 2024b) limits flexibility by enforcing preprocessed traversal routes that confine searches to neighboring nodes.

2) *How to achieve both context conciseness and search space completeness?* Concise context, such as code skeletons,

reduces noise and keeps the context manageable but risks omitting critical details for precise localization. Conversely, a fully detailed search space ensures completeness but introduces overwhelming noise, redundancy, and irrelevant exploration paths. Achieving both conciseness and completeness simultaneously is challenging, as existing methods often optimize for one at the expense of the other, leaving an open gap in effective localization.

3) *How to effectively manage context during exploration?* Large repositories often introduce noise due to ambiguities, such as function overrides and inherited classes. As the exploration process progresses, irrelevant information can accumulate, misleading the LLM and resulting in incorrect identification of bug locations. Existing frameworks (Zhang et al., 2024a; Wang et al., 2024b), merely concatenate all search results into the context, which is insufficient to manage the expanding complexity of large-scale exploration.

To address these challenges, we propose an agent system consisting of three key components:

- **Priority-Based Scheduling for LLM-Guided Actions:** To address challenge 1), we design a dynamic action scheduling system that incorporates priority queues and LLM-guided action generation for codebase exploration. The priority queue dynamically reorders actions based on their contextual relevance and urgency, solving the shortcomings of previous systems that lacked effective action management.
- **Action Decomposition with Relevance Scoring:** To resolve challenge 2), we introduce a method that decomposes high-level actions, such as class skeletons or file skeletons, into finer-grained sub-actions. These sub-actions are evaluated and ranked according to their relevance to the issue using a multi-agent workflow, ensuring comprehensive exploration while avoiding noise and redundancy.
- **Distance-Aware Searched Context Pruning:** To address challenge 3), we design a context manager that dynamically prunes the searched context. The pruning algorithm leverages a node distance heuristic within the graph-oriented codebase. By filtering out irrelevant data, the context manager ensures that exploration stays focused and aligned with the bug localization.

## 2. Related Work

### 2.1. Fault Localization Algorithms and Systems

Fault localization (FL) aims to identify suspicious locations (e.g., statements or methods) in source code that are associated with bugs. Prior to the advent of LLMs, fault localization had been extensively studied, with techniques such as spectrum-based fault localization (SBFL) (Jones & Harrold, 2005), mutation-based fault localization (MBFL) (Papadakis & Le Traon, 2015), and learning-based approaches like FLUCCS (Sohn & Yoo, 2017), DeepFL (Li et al., 2019), and TRANSFER (Meng et al., 2022). However, effective fault localization in large-scale software systems remains challenging due to the vast size of codebases and the overwhelming volume of error messages, which often exceed the capabilities of standalone learning models.

Since the advanced code and natural language understanding capabilities of LLMs, Recent studies (Yang et al., 2024a; Wu et al., 2023; Li et al., 2024; Hossain et al., 2024; Kang et al., 2023; Qin et al., 2024; Wang et al., 2024c) have proposed LLM-based FL methods. These methods incorporate agents and tools to address the challenges of large-scale systems. AUTOFL (Kang et al., 2023) enhances standalone LLMs with tool invocations, such as repository retrieval tools, for more effective exploration of code repositories. RCAgent (Wang et al., 2024c) integrates four tools (code analysis, log analysis, memory retrieval, and information collection) to support decision-making. AgentFL (Qin et al., 2024) scales LLM-based fault localization to project-level contexts by combining multiple agents with static analysis tools like Tree-sitter.

However, effectively and robustly exploring the codebase while balancing the trade-off between context granularity and search space remains a significant challenge. In contrast to existing techniques, ORCALOCA introduces a dynamic action scheduling exploration system and mechanisms to score decomposed actions, addressing these limitations effectively.

### 2.2. LLM-Agent for Software Engineering

LLMs have recently demonstrated remarkable capabilities in achieving human-level performance across a wide range of tasks, significantly advancing the field of ASE. Unlike traditional function-level or file-level coding tasks like Humaneval(Chen et al., 2021), ASE requires not only basic coding proficiency but also advanced skills in managing and interacting with code repositories. To solve such more complex tasks, LLM-based agents enhance project-level software engineering tasks by iteratively and autonomously performing actions, observing feedback, and planning future steps (Hong et al., 2023; Kong et al., 2024; Wang et al., 2024a; Yang et al., 2024b; Xia et al., 2024; Ouyang et al., 2024; Zhang et al., 2024b).

OpenHands (Wang et al., 2024b) is a community-driven platform integrating widely used agent systems to explore end-to-end LLM-based agent solutions for handling complex SE tasks. AutoCodeRover (Zhang et al., 2024b) introduces LLM agents with specialized code search methods to iteratively retrieve code context and locate bugs using test cases. Agentless (Xia et al., 2024) proposes a two-stage bug-fixing system based on a streamlined workflow approach. Repounderstander (Ma et al., 2024a) empowers agents to comprehensively understand the whole repositories by a

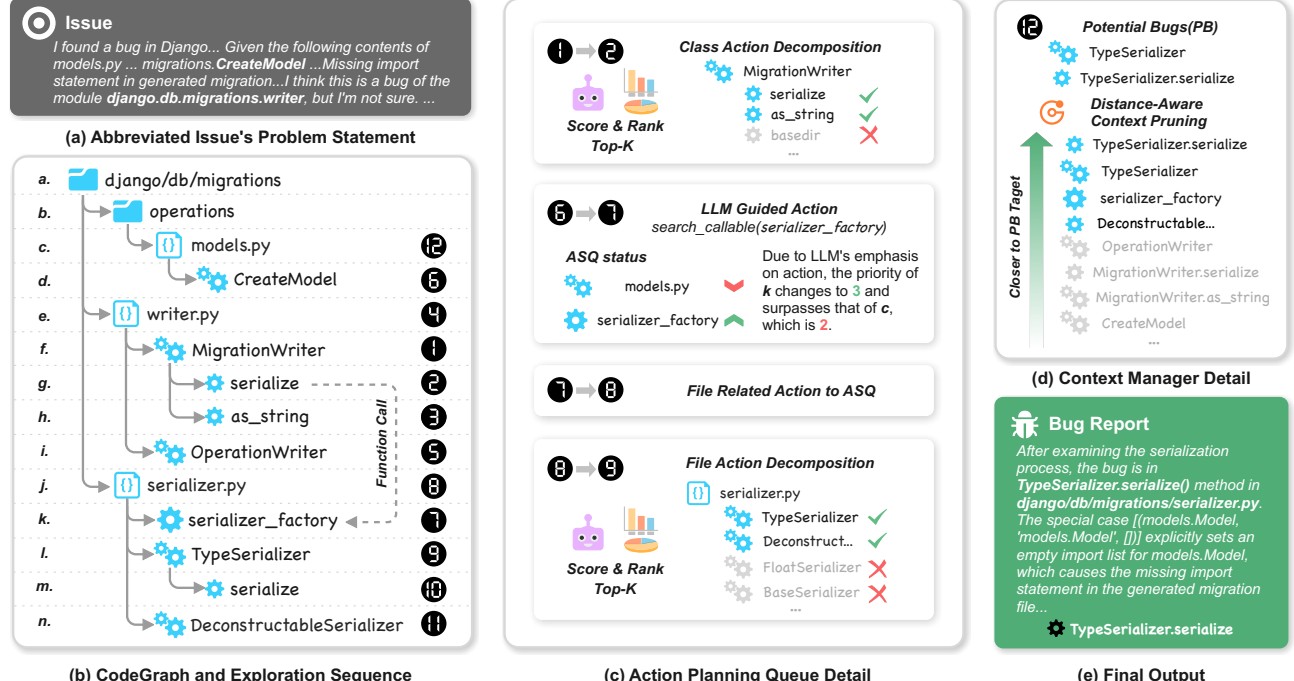

**Figure 2.** An overview of ORCALOCA using a demonstrating example from issue `django_14580`. (a) shows an abbreviated version of the issue's problem statement, where the user emphasizes `CreateModel` and `MigrationWriter`. (b) presents the exploration sequence of our agent over a part of the whole CodeGraph. (c) provides details of the Action Scheduler Queue (ASQ). Specifically, action decomposition is applied from ❶ to ❷ and from ❽ to ❾, as discussed in Section 3.3. Additionally, techniques described in Section 3.2 are used to handle steps from ❻ to ❼ and ❼ to ❽. (d) illustrates the distance-aware context pruning process, elaborated in Section 3.4. Finally, (e) shows the agent's final output. Please note this is a demonstration, experiments may use different configuration.

code knowledge graph for repositories and a Monte Carlo tree search-based repository exploration strategy.

However, existing approaches remain limited as their search processes rely entirely on the LLM to manage and guide actions, often resulting in unstable and ineffective search performance. Meanwhile, current systems, such as (Zhang et al., 2024a; Xia et al., 2024), directly incorporate all search results as context, which is inefficient and can mislead the LLM. In contrast, ORCALOCA employs a Priority-Based Action Scheduling System for LLM-guided actions and a Distance-Aware Context Pruning mechanism, significantly improving both efficiency and robustness.

## 3. Methodology

### 3.1. Search System Setup and Agent Workflow

Our search system is inspired by prior works such as (Ma et al., 2024a; Ouyang et al., 2024), which employ graph databases for indexing code repositories. Similarly, we construct a **CodeGraph**, a graph-based representation of the codebase $\mathcal{G} = (\mathcal{V}, \mathcal{E})$, to facilitate indexing and searching code entities. As illustrated in Figure 2. (b), the CodeGraph $\mathcal{G}$ contains two primary edge types $e_1, e_2 \in \mathcal{E}$. $e_1$

is containment, which represents hierarchical relationships, such as methods within classes or classes within files. $e_2$ is the reference that represents relationships such as function calls between entities. The entities include functions, classes, methods, and files. Each code entity $v \in \mathcal{V}$ in the CodeGraph is assigned with a unique identifier (UID) using the format `file_path(::cls)(::method)`. For example, in standalone functions, the UID is simply `file_path::method`. These identifiers encode the containment hierarchy directly, with `::` representing the "containment" relationship. To enhance compatibility with the **CodeGraph**, we redeveloped the API from AutoCodeRover (Zhang et al., 2024a) to provide better support for CodeGraph-based searches (See Appendix A).

Building upon the ideas of Chain of Thought (CoT) (Wei et al., 2022) and ReACT (Yao et al., 2022), ORCALOCA follows a reason-and-act workflow with a constrained action space. We design a custom-designed LLM prompt, which will generate **Observation** ($O$), **Potential Bug Locations** ($PB$), and **Search Actions** ($SA$) in each step. Here, we formulate $PB$ as a set of entities $v_{PB}$: $PB = \{v_{PB} | v_{PB} \in \mathcal{V}\}$. To better illustrate the agent workflow, we formulate it as a tuple $\mathcal{M}$, where $\mathcal{M} = (\mathcal{S}, \mathcal{C}, \mathcal{A}, \mathcal{P}, p_0)$. Here, $\mathcal{S}$ means the

state space, including previous observations, potential bug locations, and retrieved search results. $\mathcal{A}$ stands for action space, which is restricted by our search APIs. In $\mathcal{A}$, each action $a_k \in \mathcal{A}$ represents a query for retrieving relevant code snippets, generating a feedback as **Search Result** ($SR$). $\forall SR$ with UID, $SR \equiv v_{SR} \in V$. The context space $\mathcal{C}$ means for the environment, which contains the repository structure formulated by CodeGraph.

For the evolution of the agent state after action, we denote the transition function as $\mathcal{P} : \mathcal{S} \times \mathcal{A} \times \mathcal{C} \to \Delta(\mathcal{S})$. In our agent, LLM plays the key role of state transition, in which the next state $s_{t+1}$ is formed by adding new search results and refining potential bug locations. The agent follows policy $\pi : \mathcal{S} \times \mathcal{C} \to \Delta(\mathcal{A})$, which is co-managed by LLM and **Action Scheduler Queue** (ASQ). The policy determines the next action to execute based on priority, where we have a detailed description in Section 3.2. At step $t$, the action $a_t$ will also generated by the decomposition mechanism, which is described in Section 3.3.

The agent begins from the initial state $s_0$, which consists of the problem statement (See Figure 2. (a)) and the reproducer information from the issue (See Appendix C), if available. Please note that these details are concatenated in our system prompt (See Appendix D) and will be provided to LLM at each subsequent step. During the exploration, LLM agent will generate $O_t$, $PB_t$, and $SA_t$ in every step $t$. In specific, the state transition would be $O_{t+1}, PB_{t+1} \sim \mathcal{P}(O_{1...t}, SR_{1...t}^{CM})$, indicating the generated $O$ and $PB$ are dependent on all previous generated states. Here, $SR_{1...t}^{CM}$ is the pruned set of search results managed by the **Context Manager (CM)**, see Section 3.4. The process terminates when ASQ is empty or follows the convergence condition (See Appendix E). In the end, the conclusion step produces only the conclusion ($O_{\text{conclusion}}$) and the bug locations ($B$), summarizing the identified issues and their locations after all exploration steps are completed, see Figure 2. (e). Here $B = \arg\max_{PB} \mathcal{P}(PB|O_{\text{all}}, SR_{\text{all}}) \subseteq \mathcal{V}$.

Unlike traditional reinforcement learning, where the goal is to maximize cumulative rewards, our agent is designed to converge to the correct bug location effectively. The evaluation target is elaborated in Section 4.1.4.

To have a better understanding of Figure 2, we provide a core algorithm pseudocode in Algorithm 1. It summarizes the essential components discussed in Sections 3.2, 3.3, and 3.4.

For implementation details such as ASQ intial actions guided by reproducer, top-$k$ output mode, batch action execution, please refer to our discussion in Section 5.

### 3.2. Priority-Based Scheduling for LLM-Guided Actions

To solve challenge 1) we discussed in Section 1, ORCALOCA provides a more robust framework, which leverages a priority

---

**Algorithm 1** ORCALOCA Agent Core Algorithm

1: Initialize state $s_0 \leftarrow$ `problem_statement`
2: Initialize ASQ $\leftarrow \emptyset$
3: **while** ASQ not empty and not converged **do**
4:     Generate $O_t, PB_t, SA_t \leftarrow \text{LLM}(s_t)$
5:     **for all** $a_k \in SA_t$ **do**
6:         **if** $a_k$ is redundant **then**
7:             Skip $a_k$
8:         **else if** $a_k$ previously seen **then**
9:             Increment counter $C_{a_k}$ and update priority
10:         **else**
11:             Add $a_k$ to ASQ
12:         **end if**
13:     **end for**
14:     Select top-priority $a_t$ from ASQ
15:     Execute $a_t$ to get $SR_t$
16:     **if** $v_{SR} \in \mathcal{V}^{class} \vee \mathcal{V}^{file}$ **then**
17:         Generate $a_t^d$ by relevance scoring via sub-agent
18:         Add $a_t^d$ to ASQ with higher priority
19:     **end if**
20:     Pretetch SR's UID to check validity
21:     Prune $SR_{1..t}$ using CM based on distance to $PB_t$
22:     Update $s_{t+1} \leftarrow \mathcal{P}(s_t, a_t, SR_t)$
23: **end while**
24: Generate $O_{\text{conclusion}}, B \leftarrow \text{LLM}(O_{1..t}, PB_{1..t}, SR_{1..t}^{CM})$

---

queue to manage the LLM-generated actions, offering a more comprehensive and effective method for action planning.

To achieve a thorough reasoning COT, our agent limits each step to only processing one action. However, for $SA$ generated by LLM, it may have multiple action candidates based on the given context. To address this, we design a policy $\pi$ that uses a dynamic action scheduler queue (ASQ) on top of LLM-generated actions. The ASQ has priority management which is implemented on top of a heap data structure.

In ORCALOCA, action priorities are dynamically adaptable across different levels. The default priority for action $a_k \in SA$ is 1. However, this priority can be elevated based on contextual relevance and strong relationships. For instance, in Figure 2. (c), the step from ❼ to ❽ shows how the action involving the file `serializer.py` is assigned a higher priority due to its strong connection with `serializer_factory`. The same principle is set for action decomposition, which is discussed in Section 3.3.

To account for urgency, we also keep a counter $C_{a_k}$ for each unique action $a_k$. When the LLM generates the same action repeatedly, the counter $C_{a_k}$ grows, indicating the LLM's focus on checking the content. The counter $C_{a_k}$ replaces the original priority value and adjusts the position of $a_k$' in the queue. This system ensures that the most important actions are carried out first. For example in Figure 1. (c), the step

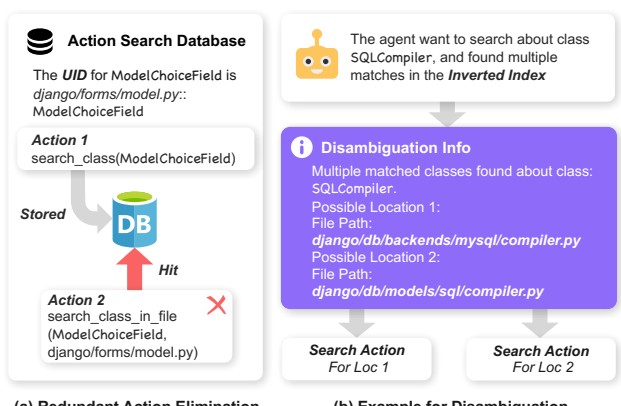

**(a) Redundant Action Elimination**     **(b) Example for Disambiguation**

*Figure 3.* Detailed examples for ORCALOCA solving redundancy and disambiguation problem.

from ❻ to ❼ shows that `serializer_factory` would come to the next step due to its counter has accumulated to 3, which even surpasses the file related action `models.py` corresponding to `CreateModel`.

Additionally, to address the unpredictability and hallucinations of LLMs, we set up a redundancy elimination mechanism to improve action scheduling. This mechanism ensures that redundant actions are avoided, enhancing efficiency and preventing unnecessary exploration.

Consider the previous agent API used by systems like (Zhang et al., 2024b; Ma et al., 2024a). When it comes to search class content, it has two different APIs `search_class(cls)` and `search_class_in_file(cls, f)` which will target at class searching. Initially, the LLM may lack precise information about the location of the target class, which leads to the use of the general method `search_class(ModelChoiceField)`. However, after analyzing the returned content, the LLM will learn the file path and generate a subsequent, more specific action, such as `search_class_in_file(ModelChoiceField, django/forms/models.py)`. Without careful handling of API ambiguities in scheduling, even a unique class like `ModelChoiceField` could result in duplicate actions and redundant content searches.

To mitigate this, as illustrated in Figure 3 (a), we maintain an action search database. Before an action is passed to the agent's chain-of-thought (COT) reasoning, we prefetch its UID from CodeGraph and register its unique identifier (UID) in this database. This prefetching process ensures that each action is checked against previously executed actions, preventing duplicates and enabling more efficient scheduling.

### 3.3. Action Decomposition with Relevance Scoring

Achieving both conciseness and completeness simultaneously is challenging. Previous solutions (Xia et al.,

2024; Zhang et al., 2024a) frequently employed skeletal techniques for huge classes or files, returning solely the class and methods signature. However, brutal traversal over all the methods could lead to noisy context and redundant actions. To overcome this challenge, we propose action decomposition with relevance scoring.

Specifically, if the search result $SR$ of an action $a_k$ corresponds to a class $v_{SR} \in \mathcal{V}^{class}$, we employ a **score and rank sub-agent** to evaluate the relevance of each method in the class $\mathcal{N}_{v^{class}} = \{v \mid v \to v^{class} \in e_1 \}$ to the problem statement. The sub-agent (implemented by another LLM agent) will select the top-$k$ most relevant methods, which are recomposed as new search actions, denoted as $a_k^d$. These decomposed actions $a_k^d$ are assigned a higher priority (e.g. 2), and pushed to the ASQ for execution. In this way, the main agent could work with the scoring sub-agent in a multi-agent workflow. Moreover, we extend this decomposition principle to handle large files. For a file that triggers skeleton mode, we collect code entities within the file, like functions and classes, and treat them as individual units for the sub-agent. We have shown the illustrated example in Figure 2. (c).

In addition to enhancing granularity, our method addresses ambiguities, which commonly appear in large software repositories such as function overrides, and inherited classes. To resolve these issues, we implement a robust disambiguation mechanism within our decomposition strategy. We first constructed an inverted index that stores only the callable indices that exhibit ambiguities. The value of the index encloses the exact location, including the file, path, and relevant class, if applicable. As shown in Figure 3. (b), when our API finds a query with ambiguities, it will locate itself in the inverted index, enabling us to gather all the possible locations to form a disambiguation message for the LLM agent. Additionally, we will split the potential locations and fine-grainedly push back the related search actions in the action queue.

### 3.4. Distance-Aware Searched Context Pruning

To prune the irrelevant context and keep LLM focusing on useful information, we developed a distance-aware context pruning method, which we call as the **Context Manager (CM)**. The CM is designed to maintain a concise and relevant set of search results ($SR$) by evaluating their relationship to the potential bug locations ($PB$).

First of all, to enhance relevance, the CM retains only $SR$ entries linked to valid search query UIDs. Disambiguation messages (See Figure 3. (b)) and skeleton messages, typically used for large files and classes, are explicitly excluded to prevent irrelevant data from polluting the context.

The pruning process is guided by CodeGraph $\mathcal{G}$, where each search result $SR$ is mapped to a unique graph node $v_{SR} \in V$. The CM evaluates each $SR$ based on

its distance to the potential bug locations $PB$, which are also represented as nodes in the graph. Specifically, the CM computes the average shortest path distance between each node $v_{SR}$ and the candidate nodes in $PB$: $d(SR, PB) = \frac{1}{|PB|} \sum_{v \in PB} \min (d(v_{SR},v), d(v, v_{SR}))$, where $d(v_{SR}, v)$ represents the shortest path from $v_{SR}$ to $v$ in the directed CodeGraph, and $d(v, v_{SR})$ represents the reverse shortest path. The final distance metric for pruning is defined as the minimum of these two values.

Once distances are calculated, the CM prioritizes the most relevant results. It selects the top-$k$ candidates based on the calculated average distance, ensuring that LLM bypass those irrelevant code blocks. As shown in Figure 2. (d), in the last step, the context will filter out the irrelevant info like `OperationWriter`, `CreateModel`, which will make the conclusion step have a stable and correct bug location output. Importantly, the CM is applied to every step during the exploration phase.

By aligning $SR$ entries with the structural relationships within the CodeGraph, the CM helps the system focus on areas most likely to contain the bug. This approach not only streamlines the input context but also improves the accuracy and efficiency of the search process.

# 4. Evaluation

## 4.1. Setup

### 4.1.1. DATASETS

**SWE-bench** (Jimenez et al., 2023) is a widely used dataset for evaluating the ability of LLM systems to address real-world software engineering challenges. It comprises 2,294 task instances derived from 12 popular Python repositories, where each task requires a patch to resolve the issue described in its corresponding GitHub issue.

To reduce evaluation costs and complexity, the SWE-bench team introduced two refined subsets:

- **SWE-bench Lite** contains 300 instances filtered using heuristics, such as removing tasks with images, external hyperlinks, or short descriptions. Each task includes functional tests to validate the correctness of submitted patches.
- **SWE-bench Verified**, developed in collaboration with OpenAI, includes 500 instances manually validated by professional annotators, providing greater reliability.

To further optimize costs for repeated experiments, we defined a smaller subset, **SWE-bench Common**, consisting of 93 instances that form the intersection of SWE-bench Lite and SWE-bench Verified. Its compact size and high reliability make it ideal for tasks such as ablation studies.

In our experiments, we evaluate the performance of ORCALOCA using SWE-bench Lite and conduct ablation studies using SWE-bench Common.

### 4.1.2. BASELINES

We compare ORCALOCA against 17 different approaches listed on the public leaderboard (Jimenez et al., 2025) of SWE-bench Lite. These approaches are categorized into 2 groups: (1) closed-source solutions, such as Alibaba Lingma (Ma et al., 2024b); (2) open-source solutions, including OpenHands (Wang et al., 2024b), AutoCodeRover (Zhang et al., 2024b), Agentless (Xia et al., 2024), RepoGraph (Ouyang et al., 2024), HyperAgent (Phan et al., 2024), and SWE-Agent (Yang et al., 2024b).

The SWE-bench Lite leaderboard mandates that each submission include the generated patches for addressing the given issues. This requirement enables the computation and comparison of a broader range of metrics beyond the resolved rate. In addition to analyzing the leaderboard data, we reproduced the Agentless-1.5 model for a direct comparison with ORCALOCA , as its editor component is integrated into our system.

### 4.1.3. IMPLEMENTATION

ORCALOCA is built on the LlamaIndex framework (Liu, 2022), which supports various foundation models. For our experiments, we used Claude-3.5-Sonnet-20241022 (Anthropic, 2024) as the underlying model, with a sampling temperature set to 0.1 to prioritize deterministic results.

For the top-$k$ values used in action decomposition (Section 3.3), we set $k = 3$ for class decomposition and $k = 2$ for file decomposition. In the context pruning (Section 3.4), the context window size is configured to retain 12 entries (top-$k$). Our framework also supports a wide range of customizable configurations, enabling users to fine-tune their agent workflows. These settings include parameters such as class decomposition, file decomposition, disambiguation decomposition, priority adjustment, and the ability to enable or customize priority levels. This flexibility allows users to tailor their agent's behavior to specific use cases, enhancing both exploration and fine-tuning capabilities. The cost of searching is about $0.87 per instance.

To evaluate the contribution of ORCALOCA to the final Resolved Rate on SWE-bench Lite, we integrated the Repair, Patch Validation, and Patch Selection components of Agentless-1.5 (Xia et al., 2024) by converting the output of ORCALOCA into Agentless format. Inspired by Repograph (Ouyang et al., 2024), the dependencies of the output code are also added. We largely adhered to the experimental setup outlined in the Agentless public repository, using the same LLM model, Claude-3.5-Sonnet-20241022. For the repair process, we generated 40 patches (1 at a temperature of 0 and the rest at 0.8) with the `str_replace_format` argument

*Table 1.* Performance and ranking on submissions of SWE-bench-Lite (See Appendix G for submission details). Cutoff: 01/13/2025. * indicates a tie in ranking. 🔒 indicated the agent is closed-source. The best results for each metric are **bolded** and labeled as 👑. The best open-source ones are underlined and labeled as ★.

† The reported results for AutoCodeRover-v2.0 were obtained from their latest submission to SWE-bench, as they did not submit to SWE-bench Lite. To ensure alignment, we manually filtered their results to match the SWE-bench Lite subset.

‡ The reported results of Agentless-1.5 are our reproduction based on the open-source code they provided. The discrepancy between this result and the one they submitted to the leaderboard could be attributed to the outdated reproduction script shared in their repository.

| LLM Agent | LLM | Resolved Rate (Count) | Rank | Function Match Rate (Count) | Rank | File Match Rate (Count) | Rank |
|---|---|---|---|---|---|---|---|
| Blackbox AI 🔒 | N/A | **49.00% (147)** | **1** 👑 | 63.33% (190) | 5 | 81.33% (244) | 6 |
| Gru (2024-12-08) 🔒 | N/A | 48.67% (146) | 2 | 61.67% (185) | 6 | 83.33% (250) | 3* |
| Globant Code Fixer 🔒 | N/A | 48.33% (145) | 3 | **67.33% (202)** | **1** 👑 | 84.00% (252) | 2 |
| devlo 🔒 | N/A | 47.33% (142) | 4 | 66.67% (200) | 2 | **84.67% (254)** | **1** 👑 |
| OpenCSG Starship 🔒 | Ⓢ GPT-4o | 39.67% (119) | 10 | 49.00% (147) | 17 | 70.67% (212) | 16 |
| Bytedance MarsCode 🔒 | N/A | 39.33% (118) | 11 | 56.33% (169) | 13 | 79.67% (239) | 7* |
| Alibaba Lingma 🔒 | N/A | 33.00% (99) | 15 | 57.33% (172) | 11 | 75.00% (225) | 13 |
| Kodu-v1 | Ⓐ Claude 3.5 Sonnet | 44.67% (134) | 5 ★ | 52.00% (156) | 15 | 65.00% (195) | 19 |
| OpenHands + CodeAct v2.1 | Ⓐ Claude 3.5 Sonnet | 41.67% (125) | 6 | 63.67% (191) | 4 | 81.67% (245) | 5 |
| PatchKitty-0.9 | Ⓐ Claude 3.5 Sonnet | 41.33% (124) | 7 | 59.67% (179) | 8 | 75.33% (226) | 12 |
| Composio SWE-Kit | Ⓐ Claude 3.5 Sonnet + Ⓢ o1-mini | 41.00% (123) | 8* | 61.00% (183) | 7 | 79.67% (239) | 7* |
| Moatless Tools | Ⓐ Claude 3.5 Sonnet | 39.00% (117) | 12 | 59.33% (178) | 9 | 79.33% (238) | 9 |
| | ⓓ DeepSeek V3 | 30.67% (92) | 16 | 54.33% (163) | 14 | 74.33% (223) | 14 |
| AutoCodeRover-v2.0† | Ⓢ GPT-4o | 37.33% (112) | 13 | 57.00% (171) | 12 | 77.67% (233) | 11 |
| Agentless-1.5‡ | Ⓐ Claude 3.5 Sonnet | 34.67% (104) | 14 | 58.67% (176) | 10 | 78.67% (236) | 10 |
| RepoGraph | Ⓢ GPT-4o | 29.67% (89) | 17 | 47.67% (143) | 18* | 70.33% (211) | 17 |
| HyperAgent | Ⓐ Claude 3.5 Sonnet | 25.33% (76) | 18 | 47.67% (143) | 18* | 67.67% (203) | 18 |
| SWE-agent | Ⓐ Claude 3.5 Sonnet | 23.00% (69) | 19 | 51.67% (155) | 16 | 71.67% (215) | 15 |
| | Ⓢ GPT-4o | 18.33% (55) | 20 | 42.00% (126) | 21 | 57.67% (173) | 21 |
| | Ⓢ GPT-4 | 18.00% (54) | 21 | 43.67% (131) | 20 | 61.00% (183) | 20 |
| | Ⓐ Claude 3 Opus | 11.67% (35) | 22 | 33.67% (101) | 22 | 47.67% (143) | 22 |
| **ORCALOCA** | Ⓐ Claude 3.5 Sonnet | 41.00% (123) | 8* | 65.33% (196) | 3 ★ | 83.33% (250) | 3* ★ |

set. During patch validation, we employed both regression and reproduction tests. Regression tests were filtered with a temperature of 0, while reproduction tests were generated using 40 samples (1 at a temperature of 0 and the rest at 0.8). Finally, the results of selected regression and reproduction tests were used to identify the most effective patch among the 40 candidates. The cost of editing is about $0.90 per instance.

### 4.1.4. METRICS

To evaluate the performance of ORCALOCA , we utilized four metrics: Resolved Rate, Function Match Rate, File Match Rate, and Function Match Precision. Each metric is designed to provide unique insights into the effectiveness and quality of the agent.

- **Resolved Rate** is a metric originally proposed by the SWE-bench benchmark (Jimenez et al., 2024), which we adopted for our evaluation. The benchmark assesses whether an issue is resolved by constructing a Docker container for each instance, applying the user-submitted patch, running regression tests within the container, and analyzing the test results. The final metric is the percentage of the instances that are resolved.

- **Function Match Rate** and **File Match Rate** assess the localization accuracy of ORCALOCA by calculating the percentage of **Match** in instances. These metrics, inspired by prior works such as Agentless (Xia et al., 2024) and Repograph (Ouyang et al., 2024), evaluate how well the agent's outputs align with the golden patch. (To align with these works, we use the term function as a general term that includes functions and methods).

  To determine **Function Match**, we define the golden and agent-generated localization function results for each instance $i$ as sets: $B_{i,\text{golden}}^{\text{func}}, B_{i,\text{agent}}^{\text{func}} \subseteq \mathcal{V}$, following definitions in Section 3.1. A match is registered if the golden set is a subset of the agent's prediction: $B_{i,\text{golden}}^{\text{func}} \subseteq B_{i,\text{agent}}^{\text{func}}$. For **File Match**, we consider the subset of file nodes in the graph $\mathcal{G}$, denoted as: $\mathcal{V}^{\text{file}}$. According the definition of our graph, every node $v \in \mathcal{V}$ is either a file node or has an ancestor by containment edge that is a file node. Thus, we define a mapping function: $\text{fileOf} : \mathcal{V} \to \mathcal{V}^{\text{file}}$, which returns the file containing node $v$. The File Match is then determined as: $B_{i,\text{golden}}^{\text{file}} \subseteq BF_{i,\text{agent}}^{\text{file}}$, where $B_i^{\text{file}} = \{\text{fileOf}(v) \mid v \in B_i\}$.

- **Function Match Precision** is a metric proposed by us

to assess the quality of localization results. For instance, a localization output that includes every function in the repository would always ensure a function match but would be practically useless. To solve this problem, the Function Match Precision is computed for each instance as $\text{FMP}_i = |B_{i,\text{golden}}^{\text{func}} \cap B_{i,\text{agent}}^{\text{func}}| / |B_{i,\text{agent}}^{\text{func}}|$, and the final metric is the average of $\text{FMP}_i$ per instances.

## 4.2. Results

### 4.2.1. PERFORMANCE ON LEADERBOARD

As shown in Table 1, our ORCALOCA sets a new open-source State-Of-The-Art (SOTA) with a Function Match Rate of 65.33% (196 out of 300) and a File Match Rate of 83.33% (250 out of 300). These results demonstrate the effectiveness of our proposed localization methodology.

Moreover, ORCALOCA demonstrates strong performance on the Resolved Rate metric, successfully resolving 41.00% (123 out of 300) issues in the SWE-bench Lite dataset. By integrating the editing capabilities of Agentless-1.5, we achieved 6.67 percentage points improvement in function match rate and 6.33 percentage points increase in the final resolved rate over its performance. These results establish ORCALOCA as a significant milestone in the research community's efforts toward developing more robust autonomous software engineering solutions.

### 4.2.2. IMPACT OF LOCALIZATION ON RESOLVED RATE

To evaluate how ORCALOCA 's improved localization enhances the final patch resolved rate, we fully reproduced Agentless-1.5 (Xia et al., 2024) on SWE-bench Lite as a baseline. As shown in Table 2, ORCALOCA outperforms Agentless-1.5 across all three key metrics: **Resolved Rate**, **Function Match Rate** and **Function Match Precision**.

Agentless-1.5 reports two sets of localization metrics due to its multi-sampling approach (four localization attempts per instance in the official reproduction). Patch generation then evenly distributes these samples, producing 10 patches per localization result (40 in total, as per Section 4.1.3). To fairly evaluate localization performance under this setting, we compute metrics using two aggregation methods:

- **Union of Locs**: Merges function sets from all localization attempts into a single aggregated union set per instance before computing metrics. This typically results in a higher Function Match Rate but a lower Function Match Precision, as more functions are included.
- **Mean of Locs**: Computes metrics separately for each localization attempt and reports the average. This method generally yields a higher Function Match Precision but a lower Function Match Rate.

As expected, the Union of Locs method captures more correct functions but also increases noise, whereas the Mean

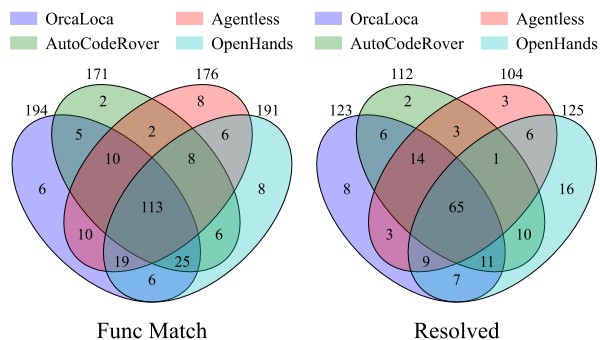

*Figure 4.* Unique localizations and solutions of open source agents.

of Locs approach filters functions more precisely at the cost of match rate.

In both cases, ORCALOCA achieves +6.67 percentage points improvement in Function Match Rate and a +4.62 percentage points increase in Function Match Precision compared to Agentless-1.5, demonstrating the effectiveness of our localization methodology. Crucially, the +6.33 percentage points gain in Resolved Rate confirms that our enhanced localization directly translates to better patch resolution.

### 4.2.3. UNIQUE LOCALIZATIONS AND SOLUTIONS

We analyze the unique issues localized and resolved by ORCALOCA compared to other open-source agents including Agentless (Xia et al., 2024), AutoCodeRover (Zhang et al., 2024b) and OpenHands (Wang et al., 2024b). As shown in Figure 4, ORCALOCA uniquely localized 6 issues, demonstrating the effectiveness of our approach. Additionally, it resolved 8 unique issues, emphasizing the impact of accurate localization in ASE. These results highlight ORCALOCA 's capability as a strong complement to other systems, even if they are developed with significantly larger resources (like OpenHands).

### 4.2.4. ABLATION STUDIES

We conducted our ablation study on SWE-bench Common, a smaller subset of SWE-bench Lite, to evaluate the contributions of each proposed method. As shown in Table 3, removing any of these methods caused a noticeable performance drop of approximately 3–5 percentage points. Specifically:

- **Priority Scheduling (Section 3.2)**: Eliminating scheduler priority weakened ORCALOCA 's heuristic planning ability, making it more susceptible to distractions from less important content.
- **File & Class / Disambiguation Decomposition (Section 3.3)**: Removing the decomposition approach restricted ORCALOCA 's ability to explore a broader search space, thereby reducing overall performance. Notice here through the experiment we prove the LLM is hard to locate with correct info by only getting the disambiguation

*Table 2.* Impact of localization on resolved rate. UL stands for Union of Locations; ML stands for Mean of Locations.

| Agent | % Resolved | Function Match | |
|---|---|---|---|
| | | Rate | Precision |
| **OrcaLoca** | **41.00%** | **65.33%** | **38.34%** |
| Agentless (UL) | 34.67% | 58.67% | 29.01% |
| Agentless (ML) | | 47.33% | 33.72% |

*Table 3.* Ablation study results. Experiment completed on SWE-bench Common dataset.

| Methods | Func. Match Rate |
|---|---|
| ORCALOCA | **76.34%** (71) |
| - w/o. priority scheduling | 73.12% (68) |
| - w/o. file & class decom. | 72.04% (67) |
| - w/o. disambiguation decom. | 70.97% (66) |
| - w/o. context pruning | 72.04% (67) |

info (See Figure 3. (b)).

- **Distance-Aware Context Pruning (Section 3.4)**: Without distance-aware context pruning, ORCALOCA was forced to handle a larger and noisier context, making it significantly more difficult to focus on the most relevant code snippet. Thus the noise will degrade the final bug localization accuracy.

## 5. Discussion

We introduce several practical extensions to our system that enhance performance, flexibility, and efficiency beyond the core workflow. In addition, we highlight current limitations and outline promising directions for future exploration.

**Initial Actions from Reproducer.** As described in Appendix C, we extract the calling stack from issue reproducers and construct first actions. This warms up the agent with more relevant context, avoiding the cold-start issue caused by relying exclusively on the problem statement and search API.

**Top-$k$ Retrieval Output Mode.** To allow customizable result granularity, we provide a top-$k$ retrieval mode, other than directly generated by agent (see Section 3.1). In this mode, the final bug location choices are chosen from the top-$k$ relevant search results ($SR$) stored by the Context Manager, allowing users to adjust the precision-recall tradeoff for various evaluation settings.

**Batch Action Execution.** To reduce reasoning length and token consumption, our system supports a batch mode where multiple top-priority actions can be executed in one step. In our experiments, we adopt a conservative batch size of 1 to maintain stable accuracy. Larger batch sizes (e.g., 2) are supported but may slightly affect the final accuracy. Thus, the batch size can be tuned depending on the desired balance between efficiency and precision.

**System Overhead and Cost Analysis.** We note the system overhead majorly introduced by dynamic code search and LLM serving. In particular, shortest-path distances on the CodeGraph are computed on the fly, and the graph itself is reconstructed per repository and commit ID to ensure semantic precision. In future work, we plan to adopt a caching mechanism to reduce redundant graph construction for frequently queried repositories. For LLM part, since it can be measured by token cost, we analyze a detailed token usage in Appendix F.

**Model Generalization.** Our experiments primarily use Claude due to its strong code reasoning capability. However, our framework is model-agnostic. Future work includes supporting open-source models such as Qwen (Yang et al., 2025) and LLaMA (Touvron et al., 2023), especially when fine-tuned and deployed locally. This extension will substantially reduce the cost of repo-level benchmark evaluations and make our approach more accessible to the community.

**Multi-Language and Cross-Language Support.** Our current implementation focuses on Python repositories, as it leverages Python-specific syntax parsing and. Supporting other languages would require integrating language-specific parsers and handling syntax and semantic differences, which introduces extra engineering overhead. Still, our general framework is language-agnostic in principle since it works on structural connections instead of language semantics. Cross-language linking presents the primary difficulty for multi-language repositories—such as Python/C++ hybrids common in ML systems. We are currently working on our index to capture inter-language relationships and enable, which will require more static analysis and more complex cross-language coordination methods.

## 6. Conclusion

We presented ORCALOCA , a framework designed to enhance software issue localization by incorporating innovative methodologies such as priority-based scheduling for LLM-generated actions, action decomposition with relevance scoring, and distance-aware context pruning to streamline the search process and improve localization accuracy. On the SWE-bench Lite benchmark, ORCALOCA achieved a 65.33% function match rate, establishing a new open-source state-of-the-art (SOTA) for software issue localization. Furthermore, by integrating the patch generation component from another open-source framework, ORCALOCA attained a final resolution rate of 41.00%, achieving a 6.33 percentage points improvement over the original framework. These contributions not only advance the field of ASE but also provide a modular framework that may inspire future research in integrating LLMs with automated debugging systems.

## Acknowledgment

This research was partially conducted using computational resources provided by the Google Cloud Platform (GCP) Credits Award.

We sincerely appreciate the valuable suggestions on paper writing provided by Yun Joon Soh and Haolan Liu from the STABLE Lab at UC San Diego.

## Impact Statement

This paper presents work whose goal is to advance the field of Machine Learning. There are many potential societal consequences of our work, none of which we feel must be specifically highlighted here.

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

# A. Code Graph Details

## A.1. Graph Construction Process

The **CodeGraph** represents the structural and semantic relationships within a codebase by integrating *containment* and *reference* relationships. It is constructed using **Abstract Syntax Tree (AST)** analysis and additional **directory-based** hierarchical relationships.

## A.2. Containment Graph Construction

The **containment graph** models the *lexical and structural hierarchy* of the codebase. We extract entities by analyzing each file in the repository using AST, identifying: **Classes**: $v^{\text{class}}$, **Functions**: $v^{\text{function}}$, **Methods**: $v^{\text{method}}$, **files**: $v^{\text{file}}$

A **containment edge** $e_1$ is added to represent hierarchical relationships: $v^{\text{method}} \to v^{\text{class}} \in e_1, \quad v^{\text{function}} \to v^{\text{file}} \in e_1$

Although directories are not code entities, we explicitly include them in the **CodeGraph** to *preserve structural context*. The directory structure is modeled as follows:

- Files within the same directory are connected via containment edges.

- A directory node is linked to its subdirectories.

- The **root directory** (`"."`) connects to all 1-depth subdirectories and files, forming the top-level hierarchy:

This could be summarized as a formula $v^{\text{file}} \to v^{\text{directory}} \in e_1, \quad v^{\text{directory}} \to v^{\text{subdirectory}} \in e_1, \quad v^{\text{directory}} \to v^{\text{root}} \in e_1$, which ensures that *file relationships and directory nesting* are explicitly represented in the **CodeGraph**.

## A.3. Reference Graph Construction

The **reference graph** captures execution dependencies between code entities, including function calls, variable references, and module imports. Using function call analysis from the AST, we add **reference edges**: $v^{\text{caller}} \to v^{\text{callee}} \in e_2$, where $e_2$ represents a **function call**. We didn't use static analysis to get references like the method A used in another function B, which we think is a future direction for better ASE.

## A.4. Heterogeneous Graph Representation

Our **CodeGraph** is a heterogeneous graph, integrating both containment relationships ($e_1$) and reference relationships ($e_2$). We efficiently apply Depth First Search (DFS) for code entity search during the agent exploration.

# B. Search API

We follow the design principle of AutoCodeRover's search API while implementing a merged design with default `file_path`. For example, in `search_class` we have default a `file_path` argument equal to None. In this scenario, we leverage LLM to decide whether it needs to add `file_path` argument or not based on the given context. To guide the agent, we provide the docstrings of the search APIs as part of the system prompt. The detailed API definition and docstring are attached below.

```python
def search_file_contents(
    self, file_name: str, directory_path: str | None = None
) -> str:
    """API to search the file skeleton
        If you want to see the structure of the file, including class and function
        ↪   signatures.
        Be sure to call search_class and search_method_in_class to get the detailed
        ↪   information.
    Args:
        file_name (str): The file name to search. Usage:
        ↪   search_file_contents("example.py"). Do not include the path, only the
        ↪   file name.
```

```
        directory_path (str): The directory path to search. Usage:
        ↪  search_file_contents("example.py", "path/to/directory")
    Returns:
        str: If file contents exceed 200 lines, we will return the file skeleton, a
        ↪  string that contains the file path and the file skeleton.
            Otherwise, we will return the file path and the file contents.
    """

def search_class(self, class_name: str, file_path: str = None) -> str:
    """API to search the class in the given repo.
    Args:
        class_name (str): The class name to search.
        file_path (str): The file path to search. If you could make sure the file
        ↪  path, please provide it to avoid ambiguity.
        Leave it as None if you are not sure about the file path.
        Usage: search_class("ModelChoiceField") or search_class("ModelChoiceField",
        ↪  "django/forms/models.py")
    Returns:
        str: The file path and the class content. If the content exceeds 100 lines,
        ↪  we will use class skeleton.
        If not found, return the error message. If multiple classes are found, return
        ↪  the disambiguation message.
        Please call search_method_in_class to get detailed information of the method
        ↪  after skeleton search.
        If the methods don't have docstrings, please make sure use
        ↪  search_method_in_class to get the method signature.
    """

def search_method_in_class(
        self, class_name: str, method_name: str, file_path: str = None
    ) -> str:
    """API to search the method of the class in the given repo.
    Don't try to use this API until you have already tried search_class to get the
    ↪  class info.
    Args:
        class_name (str): The class name to search.
        method_name (str): The method name within the class.
        file_path (str): The file path to search. If you could make sure the file
        ↪  path, please provide it to avoid ambiguity.
        Leave it as None if you are not sure about the file path.
        Usage: search_method_in_class("ModelChoiceField", "to_python") or
        ↪  search_method_in_class("ModelChoiceField", "to_python",
        ↪  "django/forms/models.py")
    Returns:
        str: The file path and the method code snippet. If not found, return the
        ↪  error message.
        If multiple methods are found, return the disambiguation message.
    """

def search_callable(self, query_name: str, file_path: str = None) -> str:
    """API to search the callable definition in the given repo.
    If you are not sure about the query type, please use this API. The query can be a
    ↪  function, class, method or global variable.
    Args:
```

```
        query_name (str): The query to search. The format should be only the name.
        file_path (str): The file path to search. If you could make sure the file
        ↪   path, please provide it to avoid ambiguity.
        Leave it as None if you are not sure about the file path.
        Usage: search_callable("ModelChoiceField") or
        ↪   search_callable("ModelChoiceField", "django/forms/models.py")
    Returns:
        str: The file path and the code snippet. If not found, return the error
        ↪   message.
        If multiple matches are found, return the disambiguation message.
    """

def search_source_code(self, file_path: str, source_code: str) -> str:
    """API to search the source code in the file. If you want to search the code
    ↪   snippet in the file.
    Args:
        file_path (str): The file path to search.
        source_code (str): The source code to search.
    Returns:
        str: The file path and the related function/class code snippet.
            If not found, return the error message.
    """
```

## C. Reproducer Agent

Although the ORCALOCA search agent can inspect and explore the code repository statically, it is unable to collect runtime information. To supplement this, we developed an auxiliary **reproducer agent** that attempts to reproduce reported issues and capture execution traces. Because successful reproduction is inherently limited (Only 38.0% of issues can be successfully reproduced in our experiment), this agent serves as a complementary analysis step rather than a core element of our search design.

As illustrated in Figure 5, the reproducer agent proceeds in three stages:

- Identifies suspicious functions and files from plain text sources, including tracebacks, code snippets, logs, and natural language descriptions;

- Reproduces the issue by generating and executing a snippet, then judges reproduction result and retries if failed;

- Extracts key information from the trace through filtering and re-ranking.

### C.1. Plain Text Parser

Extracting relevant data from execution traces is challenging due to their tremendous size. To narrow the search space, we identify initial **suspicious keywords** from the problem description.

We first segment the description into multiple patterns—tracebacks, code snippets, and natural language. Each segment is then processed using tailored prompts to extract relevant keywords with higher accuracy.

### C.2. Reproduction Snippet Generator

To reproduce the issue, we set up a conda environment inside a Docker container following the methodology in SWE-Agent (Yang et al., 2024b). We then generate and execute a reproduction snippet using an LLM and record its execution trace with VizTracer (Gao, 2025).

The snippet's output is sent to an **LLM judge**, which determines whether the issue was successfully reproduced. If successful, the reproduction log and code are forwarded to the plain text parser for further analysis.

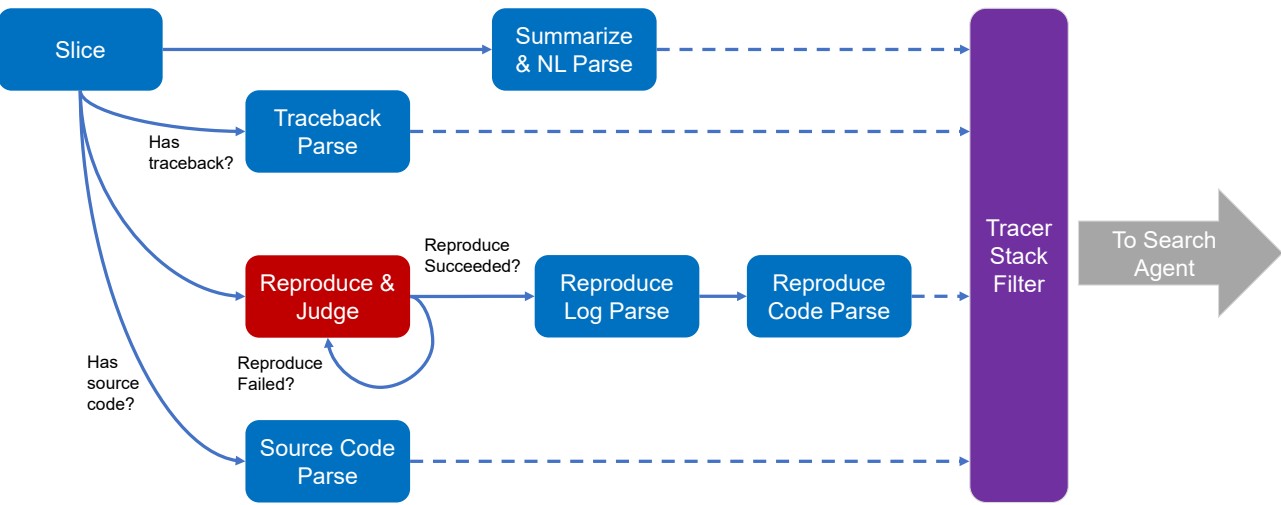

*Figure 5.* Internal structure of reproducer agent. Appendix C.1 contents are labeled in blue, C.2 in red and C.3 in purple.

## C.3. Stack Trace Selector

Once trace data is collected, we apply filtering strategies based on empirical observations. Our case study indicates that the root cause of a bug is often:

- Located in the same file as a suspicious keyword;

- A close descendant of a suspicious keyword in the trace;

- Near the root of the trace tree.

Using these heuristics, we assign priorities to trace entries and filter the top $K = 25$ candidates.

For finer-grained ranking, we compute a relevance score for each candidate by feeding its code context into an LLM. The final ranking is determined using a weighted sum of the LLM-generated score and the initial keyword-based priority. We retain candidates that exceed a predefined absolute score threshold and rank within the top 5.

## D. Key Contents in Framework Prompts

Extractor Agent Prompt

**Common System Prompt:**

```
You are an expert python developer, mastering at summarizing and extracting from
    Github issues.
```

**Slice Sub-agent:**

```
Your task is to slice strings from human reported github issue. Every slice shouldn't
    overlap with another slice.
Non-existanct slice should be set to ''.

Your output should strictly follow the format below.
{output_format}
DO NOT SPEAK ANY REDUNDANT WORDS (like 'json', 'output', etc.)
```

```
The meanings of each field are:
{output_fields}

An example is given below:
{example}

Below is the real task for you to solve:
<repo_name>{repo_name}</repo_name>
{input_description}
```

**Parse Sub-agent:**

```
Your task is to extract python code keywords and the filepath  that belong to (if exist
    ) from human reported github issue.
Non-existanct filepath should be set to ''.

Your output should strictly follow the format below.
{output_format}
DO NOT SPEAK ANY REDUNDANT WORDS (like 'json', 'output', etc.)

The meanings of each field are:
{output_fields}

An example is given below:
{example}

Below is the real task for you to solve:
<repo_name>{repo_name}</repo_name>
<input_description>
{input_description}
</input_description>
```

**Judge Sub-agent:**

```
Your task is to judge whether an input GitHub issue is successfully reproduced,
based on the reproducer_log generated by a reproducer snippet;
If the reproduce didn't succeed, try to generate a fixed reproduced snippet.

Some examples of judgment include:
1. SUCCESS if (the exact same error message) from input_description is found in
    reproducer_log;
2. FAILURE if the error message from input_description is different or irrelevant from
    the one found in reproducer_log;
3. SUCCESS if (the same printed output) from input_description is found in
    reproducer_log;
4. FAILURE if the reproducer in input_description is expected to have output (error or
    printed log) but reproducer_log is empty;
5. FAILURE if the reproducer in input_description is expected to raise an error, but no
     error is found from reproducer_log;
6. FAILURE if the reproducer in input_description is not expected to raise any errors,
    but 1 or more errors are found from reproducer_log;
7. FAILURE if the input_description describes different output for expected and
    problematic behavior, but the reproducer_log matches with the expected one;

Your output should strictly follow the format below.
{output_format}
DO NOT SPEAK ANY REDUNDANT WORDS (like 'json', 'output', etc.)
```

```
The meanings of each field are:
{output_fields}

Below is the real task for you to solve:
<repo_name>{repo_name}</repo_name>
<input_description>
{input_description}
</input_description>
<reproducer_snippet>
{reproducer_snippet}
</reproducer_snippet>
<reproducer_log>
{reproducer_log}
</reproducer_log>
```

**Summarize Sub-agent:**

```
Your task is to summarize a human-reported GitHub issue in natural language.

Your output should strictly follow the format below.
{output_format}
DO NOT SPEAK ANY REDUNDANT WORDS (like 'json', 'output', etc.)

The meanings of each field are:
{output_fields}

An example is given below:
{example}

Below is the issue for you to summarize:
<repo_name>{repo_name}</repo_name>
<input_description>
{input_description}
</input_description>
```

**Code Scorer Sub-agent:**

```
You are a Python coding expert. Your job is to score how likely a piece of code will
    need to be modified to solve a GitHub issue. The issue description will be
    presented in 'problem_statement'.

<problem_statement>
{problem_statement}
</problem_statement>

Please score how likely this piece of code will need to be modified to solve a GitHub
    issue. Please score the likeliness with an integer between 0 and 100, the higher
    the more likely. Your output will be processed by a program instead of a human, so
    please ONLY output a single integer.
```

Searcher Agent Prompt

```
You are a professional software engineer who uses API calls to report bug code snippets
     from a text into json format.
You need to extract where are the bug locations by analyzing the text.
The given text contains the problem statement and the code snippets.
```

```
There are some API calls that you can use to extract the information.
The API calls include:
{tool_desc}

<TASKS>
Every time you will do the following things:

1. Provide the observation based on given input:
Every time we will provide a new search result in tag <New Info>.
It may contain the disambiguation info if the search action is related to multiple
    classes or methods.
Also, previous search results will be provided in the tag <Search Result>. You need to
    analyze the new search result based on the previous one and provide the observation
based on the whole context.
2. Think about where the bug might be in the code by the whole given context(including
    all Search Result), and provide the potential bug locations. The potential here
    means the most possible locations up to the current context.
3. Check whether it contains any class, method, or function you need to further search.
     Especially, if disambiguation info is provided, you need to search for the
    specific class or method.
Plan the new_search_actions based on the current context. You can use the given API
    calls to search for the bug locations.
You can put multiple actions in the new_search_actions list. Be sure to use arguments
    in the tool description.
If you make sure the context is enough to answer the question, you can keep the
    new_search_actions list empty.

The conclusion is a final standalone step to provide the final bug locations when
    nothing else to search. Please keep in mind to
follow the instruction "Now let's come to a conclusion. ".
</TASKS>

<OUTPUT FORMAT>
1. Regular Step Format:
   Provide your answer in a clear JSON structure like this,
   {step_format}
   Make sure each API call is written as a valid Python expression and code_snippet is
    a valid Python string.
   In potential_bug_locations, you should provide the file path, class name, and
   method name.
   It's not the final answer, just a hint for possible bug locations.
   If the method does not belong to any class, set the class to an empty string.
   You can provide multiple actions in the new_search_actions. DO NOT add any title or
    description.
2. Conclusion Format:
   After no input actions in the search queue, provide the final bug locations in JSON
    structure like this.

   {bug_locations}
   DO NOT generate observation or new_search_actions in the conclusion step.
   DO NOT mix it with any title or description. If the method does not belong to any
   class, set the class to an empty string.
</OUTPUT FORMAT>
```

## E. Convergence Configuration

Early Stop Convergence Mode In most cases, our agent naturally converges when there are no remaining actions in ASQ. However, in scenarios where the action sequence is lengthy and requires multiple execution steps, we introduce an early stop convergence mode to optimize efficiency.

This mode is controlled by a BERT embedding model, which evaluates the similarity between consecutive observations at

each step. Specifically, for two observations, $O_t$ and $O_{t+1}$, we compute their cosine similarity using their BERT embeddings:

$$\cos\theta = \frac{\langle \text{BERT}(O_t), \text{BERT}(O_{t+1}) \rangle}{|\text{BERT}(O_t)| \cdot |\text{BERT}(O_{t+1})|}$$

If the similarity score exceeds 0.97, the two observations are considered equivalent.

To ensure stability in the decision-making process, we apply a sliding window mechanism over consecutive observations. Specifically, we require that the similarity condition holds for $K = 15$ consecutive steps before triggering convergence:

$$\sum_{i=t}^{t+K-1} \mathbb{1}(\cos\theta_i > 0.97) = K$$

Once this condition is met, the agent terminates execution and reaches a conclusion.

## F. Cost Breakdown Analysis

We chose token cost as our primary metric because LLM inference dominates the overall time and monetary expenses of our system. For runtime analysis, since our implementation primarily leverages API services from external model providers, inference time can be considered approximately proportional to token usage.

In Table 4, we summarize the average per-instance token cost across different agents:

*Table 4.* Average token cost per instance for different agents.

| Agent | Cost |
| --- | --- |
| OpenHands | 1.14 |
| SWE-Agent | 1.62 |
| AutoCodeRover | 1.30 |
| Agentless-1.5 | 1.05 |
| OrcaLoca | 1.77 |
| OrcaLoca-batch(=2) | 1.48 |

Notably, over half of OrcaLoca's token cost is attributed to the editing phase (0.90 out of 1.77), which is primarily contributed by the edit component from Agentless-1.5, as we adopt their editing mechanism in our implementation. Although this paper primarily targets performance and accuracy, the reduced cost observed in OrcaLoca-batch highlights a large optimization potential for improving efficiency in the localization phase.

In OrcaLoca-batch, we implemented batched action execution during localization, extracting the top-priority actions in groups from the scheduler (See Section 5. Table 5 presents a comparison of old and new token costs across ten sampled issues from SWEBench-Lite. The ratio (New Cost / Old Cost) reflects the cost improvement:

*Table 5.* Token cost comparison before and after batched action optimization.

| Instance ID | Old Cost | New Cost | Ratio |
|---|---|---|---|
| django-13551 | 0.30 | 0.26 | 0.87 |
| django-15814 | 1.44 | 0.97 | 0.67 |
| django-16255 | 0.17 | 0.18 | 1.06 |
| pylint-7228 | 0.71 | 0.66 | 0.93 |
| pytest-8906 | 1.93 | 0.87 | 0.45 |
| scikit-learn-13439 | 0.31 | 0.21 | 0.68 |
| sympy-14774 | 0.53 | 0.15 | 0.28 |
| sympy-15011 | 1.14 | 0.64 | 0.56 |
| sympy-16792 | 1.05 | 0.64 | 0.61 |
| sympy-24213 | 0.55 | 0.20 | 0.36 |

Due to budget constraints, we sampled 10 issues with varied token profiles. Using weighted averages across cost bins, we estimate that per-instance localization cost was reduced by an average of 34% (from 0.87 to 0.58) without negatively impacting localization correctness.

We are committed to further optimizing OrcaLoca and plan to explore additional efficiency improvements in future work, such as integrating `kv-cache` techniques during inference.

## G. Other Competing Methods

- **Blackbox AI Agent** (Blackbox, 2024) is building coding agent to transform the way we build software.

- **Gru(2024-12-08)** (Gru, 2024) builds different agents to solve different software engineering problems. But all Grus are built with the same principles: Clear Problem Domain, Dedicated Tools and Direct Value Delivery.

- **Globant Code Fixer Agent** (Globant, 2024) is an independent and intelligent software entities designed to transform business operations.

- **devlo** (devlo, 2024) boosts user's productivity by handling development tasks, freeing user to focus on innovation and ship products faster.

- **OpenCSG Starship Agentic Coder** (OpenCSG, 2024) is a multi-agent collaborative and scalable environment to empower user in building the next generation of intelligent applications.

- **Bytedance MarsCode Agent** (Liu et al., 2024) is a novel framework that leverages LLMs to automatically identify and repair bugs in software code.

- **Alibaba Lingma Agent** (Ma et al., 2024b) understands the whole software repository to achieving automatic software engineering.

- **Kodu-v1** (Kodu-AI, 2024) implements a VS Code extension that adapts to user's skill level, helping user bring ideas to life faster than ever before.

- **OpenHands + CodeAct v2.1** (Wang et al., 2024b) is a platform for the development of powerful and flexible AI agents that interact with the world in similar ways to those of a human developer: by writing code, interacting with a command line, and browsing the web.

- **PatchKitty-0.9**: It may have been developed concurrently with our work and is reportedly designed by researchers from UC Santa Barbara. While it was claimed to be open-source in its SWE-bench Lite submission, no repository or related links have been released yet.

- **Composio SWE-Kit (2024-10-30)** (Composio, 2024) helps user connect AI agents to external tools like Gmail, GitHub, Salesforce, etc. It's like a bridge between user's AI and the tools it needs to get work done.

- **Moatless Tools** (Moatless, 2024) is a hobby project where the authors experiment with some ideas they have about how LLMs can be used to edit code in large existing codebases. They believe that rather than relying on an agent to reason its way to a solution, it is crucial to build good tools to insert the right context into the prompt and handle the response.

- **AutoCodeRover-v2.0** (Zhang et al., 2024b) is an automated approach for solving Github issues to autonomously achieve program improvement, where LLMs are combined with sophisticated code search capabilities, ultimately leading to a program modification or patch.

- **Agentless-1.5** (Xia et al., 2024) is an agentless approach to automatically resolve software development issues. Compared to the verbose and complex setup of agent-based approaches, it employs a simplistic three-phase process of localization, repair, and patch validation, without letting the LLM decide future actions or operate with complex tools.

- **RepoGraph** (Ouyang et al., 2024) is a plug-in module that manages a repository-level structure for modern AI software engineering solutions.

- **HyperAgent** (Phan et al., 2024) is a novel generalist multi-agent system that addresses a broad spectrum of SE tasks across multiple programming languages by emulating the workflows of human developers.

- **SWE-agent** (Yang et al., 2024b): is a system that facilitates LM agents to autonomously use computers to solve software engineering tasks. SWE-agent's custom agent-computer interface (ACI) significantly enhances an agent's ability to create and edit code files, navigate entire repositories, and execute tests and other programs.

