# OpenReview forum: "OrcaLoca: An LLM Agent Framework for Software Issue Localization"
_ICML.cc/2025/Conference — ICML 2025 poster_

### Official Review · Reviewer_9wos · 2025-03-09

**Overall Recommendation:** 4

**Summary:**

This paper proposes a novel framework for automated software issue localization tasks. Issue localization is a critical component of autonomous software engineering, making the problem addressed in this paper well-motivated. The proposed technique combines several strategies to improve localization accuracy and efficiency:
- Use of a priority queue to identify higher-relevance or more urgent actions first.
- Decomposition of search actions for finer-grained and more relevant operations.
- Preservation of the most relevant code snippets as the agent iteratively retrieves them using a shortest-path heuristic on the code graph (a graph-based representation of the codebase that facilitates indexing and efficient search while capturing both code containment and references).

Experiments are conducted on the widely used SWE-bench Lite benchmark for Python projects. The authors report:
- A new open-source state-of-the-art Function Match Rate of 65.33% (an approximately 2% improvement over the previous open-source best).
- A File Match Rate of 83.33%.
- A final Resolved Rate (percentage of issues successfully fixed by the generated patches) of 41.00%.

Additionally, an ablation study is conducted on a smaller subset of SWE-bench, demonstrating that each component of the proposed approach positively contributes to the overall improvements.

## update after rebuttal
Thank you for your response to my questions. I don’t have any further questions at this time. After considering all of the discussions here, I’ve decided to keep my original score.

**Claims And Evidence:**

Overall, the paper’s main claims are well-supported by experimental results. Specifically, it provides detailed evidence for the following:
- OrcaLoca achieves a new state-of-the-art function match rate among open-source systems.
- OrcaLoca improves the final resolved rate by 6.33 percentage points over a baseline system.
- Priority-based scheduling and action decomposition enhance relevance and reduce noise in context.

**Essential References Not Discussed:**

In general, the references from the emerging field of LLM-based bug fixing are fairly comprehensive.

**Experimental Designs Or Analyses:**

OrcaLoca is tested on the SWE-bench Lite, a widely used curated set of 300 real-world GitHub issues. The authors also conduct an ablation study on a smaller subset, SWE-bench Common. They compare with 17 solutions (both closed-source and open-source). I think this is fairly comprehensive evaluation of their technique. The paper also systematically removes each of the major components to gauge its importance on localization performance.

**Methods And Evaluation Criteria:**

The authors use standard SWE-bench metrics (Resolved Rate, Function Match Rate, and File Match Rate) along with a new metric, Function Match Precision, to measure localization accuracy. This approach is reasonable and aligns with established practices in the emerging field of LLM-based bug fixing.

**Other Comments Or Suggestions:**

It would be interesting to see a break-down of performance across different types of issues (e.g., small vs. large repositories, etc.).

**Other Strengths And Weaknesses:**

Strengths:
- The three major components (priority scheduling, decomposition, distance-based pruning) are distinct yet cohesive ideas. I think the paper is giving future researchers a clear blueprint for adaptation or replication.
- Strong empirical results: SWE-bench Lite is a practical, recognized dataset, making the results fairly credible. The jump in function match rate and final resolved rate, plus ablation tests, provide strong evidence that each piece is adding value.

Weaknesses:
- While the overall approach is innovative, parts of the paper are difficult to parse. Section 3.1, in particular, introduces multiple details in quick succession. More introductory context would help guide readers through the agent workflow. Additionally, the paper references external APIs from prior systems (e.g., “Consider the previous agent API used by systems like (Zhang et al., 2024b; Ma et al., 2024a)”) without summarizing them, which may confuse less-informed readers. I encourage the authors to use available space more effectively to provide sufficient grounding before diving into technical details. Ensuring the paper is self-contained and introducing the agent framework more gradually would significantly improve readability.
- The approach is tested entirely on Python issues. It’s unclear how well the code-graph approach generalizes to, say, Java projects or cross-language scenarios.

**Questions For Authors:**

- Q1: How do you envision OrcaLoca’s performance scaling for extremely large repositories (e.g., tens of thousands of files, multi-million-line codebases)?
- Q2: Do you anticipate any major challenges when applying OrcaLoca to other programming languages or multi-language repositories?
- Q3: Your approach relies on shortest paths in a code graph. For large repositories, do you cache or approximate these distances, or are they computed dynamically each time?

**Relation To Broader Scientific Literature:**

The related works section of the paper is fairly comprehensive and includes sufficient details on prior LLM-based fault localization and debugging frameworks (including Agentless, AutoCodeRover, Repograph, as well as spectrum-based and mutation-based approaches for debugging). I agree with the authors that the main novelty of this paper is the application of multi-step agent designs to codebase-level tasks. The approach successfully bridges the gap between large LLM text-generation capabilities and more “surgical” code exploration techniques.

**Theoretical Claims:**

The paper does not present any formal theoretical claims or proofs.

---

> ### Author Rebuttal · Authors · 2025-03-31
>
> We thank the reviewers for their thoughtful feedback and for highlighting several key strengths of our work. We listed our responses for weaknesses and questions below:
>
> **Weakness 1:** While the overall approach is innovative, parts of the paper are difficult to parse... Ensuring the paper is self-contained and introducing the agent framework more gradually would significantly improve readability.
>
> **Response:**
> Thank you for your useful advice. We agree that Section 3.1 contains a lot of technical information in a small space, which may hinder readability. In the revision, we will summarize the APIs utilized in previous systems (Zhang et al., 2024b; Ma et al., 2024a) to make the paper more self-contained and accessible to a wider audience.
>
> **Weakness 2:** The approach is tested entirely on Python issues. It’s unclear how well the code-graph approach generalizes to Java or cross-language settings.
>
> **Response:**
> This is an important point. Currently, we rely on SWE-Bench, which is the state-of-the-art benchmark so far for evaluating software agents and is Python-based. We recently found more new benchmarks like SWE-Gym and LocBench, but these are also Python-based. We recognize the need for broader generalization and are actively working on developing a cross-language benchmark to evaluate agent performance in more diverse environments. Our graph-based framework could be improved further to support other languages with appropriate parsing and indexing backends.
>
> **Q1:** How do you envision OrcaLoca’s performance scaling for extremely large repositories (e.g., tens of thousands of files, multi-million-line codebases)?
>
> **Response:**
> This is an excellent question. Scaling to extremely large repositories such as Linux or Android codebases is an exciting but challenging direction. Key bottlenecks include efficient index management and the context management of CoT (Chain-of-Thought) reasoning during retrieval. In future work, we aim to incorporate an efficient Process Reward Model (PRM) to compress reasoning chains while reducing overall token consumption. We are also exploring hierarchical indexing and chunking strategies to handle large codebases more efficiently.
>
> **Q2:** Do you anticipate any major challenges when applying OrcaLoca to other programming languages or multi-language repositories?
>
> **Response:**
> Applying OrcaLoca to other single-language repositories generally presents no fundamental issues, as our system operates on the code structure and relationships. However, languages with strong type systems (e.g., C++, TypeScript) offer opportunities for type inference integration, which could enhance LLM reasoning by leveraging compiler-level information. This could reduce reasoning effort and improve localization accuracy.
>
> For multi-language repositories—especially common in ML systems (e.g., Python + C++ via PyBind)—the main challenge lies in cross-language linking. We are currently extending our index to infer connections across language boundaries and enhance reasoning with cross-language semantic context. This will require both richer static analysis and more advanced reasoning strategies.
>
> **Q3:** Your approach relies on the shortest paths in a code graph. Are these distances cached or computed dynamically?
>
> **Response:**
> Currently, shortest-path distances are computed dynamically, as our current system serves primarily as a proof of concept. That said, we recognize the performance bottleneck this can introduce. In future iterations, we plan to add a caching layer and incremental computation mechanisms to improve efficiency, especially for large or frequently accessed repositories.

---

> > ### Comment · Reviewer_9wos · 2025-04-08
> >
> > Thank you for your response to my questions. I don’t have any further questions at this time. After considering all of the discussions here, I’ve decided to keep my original score and recommend acceptance of the paper.

---

### Official Review · Reviewer_BMAE · 2025-03-11

**Overall Recommendation:** 3

**Summary:**

This paper introduces OrcaLoca, an LLM agent framework that improves software issue localization by integrating three key components: priority-based scheduling for LLM-guided actions, action decomposition with relevance scoring, and distance-aware context pruning. Experimental results demonstrate that OrcaLoca achieves good performance on the SWE-bench Lite benchmark with a function match rate of 65.33% and a file match rate of 83.33%. Additionally, by integrating patch generation capabilities, OrcaLoca improves the resolved rate by 6.33 percentage points compared to existing frameworks.

**Claims And Evidence:**

The claims in the paper are well-supported by comprehensive experimental evidence. The authors provide detailed performance metrics on the SWE-bench Lite benchmark, comparing OrcaLoca against 17 different approaches.

**Essential References Not Discussed:**

The paper covers most essential references in the field.

**Experimental Designs Or Analyses:**

I carefully examined the experimental designs and analyses presented in the OrcaLoca paper. Overall, the experimental methodology appears sound, with appropriate benchmarks, metrics, and comparative analyses.

However, **there are also some limitations**: more in-depth analysis of ablation experiments, evaluation on multiple models, and evaluation of costs compared to other methods.

**Methods And Evaluation Criteria:**

The evaluation methodology is sound and appropriate for the problem. The authors use established benchmarks (SWE-bench Lite and SWE-bench Verified) and metrics (Resolved Rate, Function Match Rate and File Match Rate) that are standard in the field.

**Other Comments Or Suggestions:**

A more in-depth analysis of **complex components** would be helpful to improve this paper.

**Other Strengths And Weaknesses:**

**Strengths**
- The paper addresses a significant challenge in Autonomous Software Engineering (ASE) with a novel approach that demonstrably outperforms existing methods.
- The framework design is comprehensive, tackling multiple identified challenges in LLM-based localization.
- The analysis of unique localizations (Figure 4) demonstrates OrcaLoca's ability to solve issues that other systems cannot.

**Weaknesses**

- **Complexity of the approach**: While the ablation studies show that each component contributes to performance, the overall design is complex with multiple interacting parts. The results show that removing individual components reduces performance by 3-5 percentage points, which raises questions about whether a simpler design might achieve comparable results with less complexity.

- **Computational overhead**: The paper doesn't thoroughly discuss the computational costs associated with OrcaLoca's approach. While Section 4.1.3 mentions that "the cost of searching is about 0.87, the cost of editing is about 0.90 per instance," a more detailed analysis of time and resource requirements compared to other approaches would strengthen the paper.

- **Model dependency**: The experiments primarily use Claude-3.5-Sonnet-20241022, and it's unclear how the approach would perform with other widely used models like GPT-4o or open-source models. Testing with a broader range of LLMs would demonstrate the generalizability of the approach.

**Questions For Authors:**

See about "Other Strengths And Weaknesses".

**Relation To Broader Scientific Literature:**

The paper thoroughly discusses related work in LLM-based fault localization, positioning OrcaLoca's contributions in the context of existing approaches. The authors clearly identify limitations in previous systems such as Zhang et al. (2024a), Wang et al. (2024b), and Agentless (Xia et al., 2024), explaining how OrcaLoca addresses these limitations.

**Theoretical Claims:**

The paper does not make formal theoretical claims requiring proof verification.

---

> ### Author Rebuttal · Authors · 2025-03-31
>
> We thank the reviewers for their thoughtful and constructive feedback, as well as for highlighting several key strengths of our work. For the responses for weakness, we listed our analysis part by part below:
>
> **Weakness 1:** The complexity of the approach
>
> **Response:**
>
> Thank you for your suggestions. We are actively exploring strategies to simplify the workflow while maintaining localization performance. One promising direction involves using a reward model to efficiently pre-score relevant code indices in a single pass, followed by a more powerful LLM to process only the high-value candidates. However, these methods are still under experiment. The current area is relatively new so there exists a big design space to explore and have adventure. We are also very interested to see if there exists a more elegant and efficient localization method.
>
>
> **Weakness 2:** Computational overhead
>
> **Response:**
>
> Thank you for highlighting the importance of a detailed computational cost analysis. We agree that comparing resource usage across approaches is valuable for readers.
>
> We chose token cost as our metric because LLM inference dominates our system's overall time and money expenses. For time analysis, since we are majorly leveraging the API service from model providers, it could seen as a metric proportional to the token count.
> In our revised manuscript, we will include a table summarizing the token cost for each agent:
>
> | Agent          | Cost |
> | ---|---|
> |      OpenHands | 1.14 |
> |      SWE-Agent | 1.62 |
> |  AutoCodeRover |  1.3 |
> |      Agentless | 1.05 |
> |       OrcaLoca | 1.77 |
> | OrcaLoca-batch | 1.48 |
>
> Notably, over half of OrcaLoca's cost is attributed to the editing phase (0.90 out of 1.77), which is influenced by the specific edit tool used.
> Although we majorly target performance and accuracy issues in our paper, localization has a large potential for efficiency optimization shown by OrcaLoca-batch.
>
> In OrcaLoca-batch, we implemented a batched actions optimization for the localization process, where we extracted the top batch steps from the priority queue.
> The table below summarizes the old and new token costs per instance, with the ratio (New Cost / Old Cost) indicating the improvement:
>
> | Inst ID            | Old Cost | New Cost | Ratio |
> | ---|---|---|---|
> |       django-13551 |      0.3 |     0.26 |  0.87 |
> |       django-15814 |     1.44 |     0.97 |  0.67 |
> |       django-16255 |     0.17 |     0.18 |  1.06 |
> |        pylint-7228 |     0.71 |     0.66 |  0.93 |
> |        pytest-8906 |     1.93 |     0.87 |  0.45 |
> | scikit-learn-13439 |     0.31 |     0.21 |  0.68 |
> |        sympy-14774 |     0.53 |     0.15 |  0.28 |
> |        sympy-15011 |     1.14 |     0.64 |  0.56 |
> |        sympy-16792 |     1.05 |     0.64 |  0.61 |
> |        sympy-24213 |     0.55 |      0.2 |  0.36 |
>
> Due to the cost budget of experiments, we sampled 10 issues from the original SWEBench-Lite with different levels of token cost. Using weighted averages based on sampled instances in each cost bin, we estimate that the per-instance cost for localization was reduced by an average of 34% (from 0.87 to 0.58) with no adverse effect on localization correctness.
>
> We are also committed to further refining OrcaLoca and will explore additional optimizations (e.g., implementing kv-cache techniques) in future work.
>
> **Weakness 3:** Model dependency
>
> **Response:**
>
> We agree that model generalization is an important concern. In our experiments, we have actually tested with models like GPT-4o and Gemini 2.0, and they worked well in our test samples, proving our agent framework is model-agnostic.  However due to budget constraints, we could not exhaustively evaluate across a wider range of LLMs. We majorly tested Claude because it was the best model with code reasonability at that time. In future work, we plan to include open-source models such as Qwen, LLaMA, and others—particularly once we fine-tune and serve our models locally, which will largely save cost for the repo-level benchmark evaluation.

---

> > ### Comment · Reviewer_BMAE · 2025-04-02
> >
> > Thank you for the author's reply, I will maintain my positive score.

---

### Official Review · Reviewer_BLXs · 2025-03-13

**Overall Recommendation:** 4

**Summary:**

The authors presented a new agentic framework called OrcaLoca to identify relevant code and resolve issues in software engineering problems. They introduce 3 new approaches: giving each action a relevancy score, maintaining a priority queue of actions where they are dynamically reordered based on contextual relevance and a context manager which prunes the search context.
These methods ensure that actions closely aligned with the issue are executed first resulting in more focused search. The context manager further helps in actively filtering out irrelevant context, reducing noise. Overall, this framework offers a precise approach to software issue localization compare, making it valualbe for finding and fixing issues in large scale codebase.

**Claims And Evidence:**

Yes

**Essential References Not Discussed:**

NA

**Experimental Designs Or Analyses:**

Yes.

**Methods And Evaluation Criteria:**

Yes

**Other Comments Or Suggestions:**

NA

**Other Strengths And Weaknesses:**

Strengths
- The paper presents novel approaches to address the problems caused by static action prediction from LLMs, especially when search space is large. This could be translated to other agentic framework such as searching from large unstructured databases.

Weaknesses
- Although their appraoch leads to SOTA performance on Function and File Match, i.e., software issue localization problem, these improvements do not translate into the highest Resolved rate. The authors should comment on this discrepancy. It would be valuable to understand why method like Kodu-v1 achieved higher Resolved rate despite having significantly lower Function Match rate.

**Questions For Authors:**

NA

**Relation To Broader Scientific Literature:**

Most agentic workflows involve adding reasoning, few shot examples and improving retrieval for improving performance. However, they actions are usually static, and thus explorations can suffer from noise accumulation. OrcaLoca improves on that idea by introducing
 relevancy score to actions, priority scheduling mechanism and context pruning techniques - improving overall search.

**Theoretical Claims:**

NA

---

> ### Author Rebuttal · Authors · 2025-03-31
>
> We thank the reviewers for their thoughtful and constructive feedback, as well as for highlighting several key strengths of our work. We also thank you for raising this important point regarding the discrepancy between Function/File Match performance and the Resolved rate in baselines.
>
> Generally speaking, higher localization performance usually offers a better background for the stage of downstream editing (Fig.1). However, since the editors are different across different implementations, the final resolved rate is not guaranteed to be high by providing a higher localization rate. For instance, although HyperAgent and RepoGraph agents scored exactly the same in the Function Match in Table.1, HyperAgent's inadequate editing capability results in a final resolved rate less than that of HyperAgent.
>
> Another reason is caused by agent workflow and algorithm stage plan. In modular systems where localization is seen as a separate step, localization provides an exact starting point that can greatly improve the quality of the final changes, which is shown in Table 2. However, not all agents follow a strictly modular localization → edit pipeline. Some approaches, such as swe-search[1], interleave editing, and localization stages, where editing feedback can influence and refine the localization in an iterative loop. In such designs, the final Resolved rate may align more directly with localization performance because the two components are tightly coupled. Kodu-v1 may also have particular algorithms to set up a tighter link between localization and edit stages.
>
> [1] SWE-Search: Enhancing Software Agents with Monte Carlo Tree Search and Iterative Refinement
> https://arxiv.org/abs/2410.20285
>
> Note: We discovered that we omitted to mention this study since it was under the Moatless tool and not reported on SWE-Bench. Later on, we will include this in our revised version.

---

> > ### Comment · Reviewer_BLXs · 2025-04-02
> >
> > Thank you for addressing my concerns. I’ll keep my rating at 4.

---

### Official Review · Reviewer_LnLT · 2025-03-14

**Overall Recommendation:** 3

**Summary:**

OrcaLoca is a novel framework that leverages LLM agents to improve software issue localization by precisely identifying the problematic sections within large codebases. The paper introduces three key innovations: priority-based scheduling for dynamically managing LLM-guided actions, action decomposition with relevance scoring to break down high-level queries into more granular sub-actions, and distance-aware context pruning that filters out irrelevant code to maintain focus. These techniques address longstanding challenges in automated bug localization, including imprecise navigation and excessive context noise, by integrating a structured code graph representation with targeted search strategies. Empirical evaluations on the SWE-bench Lite dataset demonstrate that OrcaLoca not only achieves SOTA performance with a 65.33% function match rate and 41.00% resolution rate.

**Claims And Evidence:**

1. The claim regarding a 6.33% improvement in the final resolved rate due to integration with a patch generation component is quantitatively supported. However, this improvement is context-dependent, relying on the successful integration of another system, which might not generalize across different systems.

**Essential References Not Discussed:**

N/A

**Experimental Designs Or Analyses:**

1. Although the experiments use a low-temperature setting (0.1) to promote deterministic behavior, the inherent variability of LLM outputs might still introduce some fluctuations in performance, which are not deeply discussed.

2. The performance improvements, particularly the increase in the final resolved rate, partly depend on the integration with an external patch generation component. This dependency may affect the reproducibility of the results if similar integration conditions are not met.

**Methods And Evaluation Criteria:**

Both the methods and the evaluation criteria are thoughtfully chosen and make strong sense for the application.

**Other Comments Or Suggestions:**

N/A

**Other Strengths And Weaknesses:**

Strengths:
1. The introduction of a priority-based scheduling mechanism for LLM-guided actions enhances the strategic exploration of large codebases.
2. Decomposing high-level actions with relevance scoring allows for more precise localization by breaking down complex tasks into manageable sub-actions.
3. The distance-aware context pruning method minimizes noise by filtering out irrelevant code, thereby maintaining focus on pertinent sections.


Weaknesses:
1. Heavy reliance on LLM outputs can lead to unpredictability and occasional hallucinations, affecting consistency in localization results. For example, different LLMs or even different LLM versions might yield different results.
2. The evaluation primarily focuses on SWE-bench datasets.
3. The paper provides limited discussion on computational overhead and scalability, leaving questions about resource requirements for practical deployment.

**Questions For Authors:**

N/A

**Relation To Broader Scientific Literature:**

The paper extends existing fault localization and automated debugging techniques by integrating LLM-based methods with graph representations and hierarchical planning, drawing on ideas like CoT reasoning and repository-level code graphs. By benchmarking on SWE-bench datasets and comparing with systems such as AutoCodeRover and Agentless, the work demonstrates how its contributions address limitations in prior research while paving the way for more precise LLM-driven code search and context management.

**Theoretical Claims:**

The paper primarily focuses on presenting a novel framework with empirical supports.

---

> ### Author Rebuttal · Authors · 2025-03-31
>
> We thank the reviewers for their thoughtful feedback and for highlighting several key strengths of our work. For the weak points, we list our responses part by part, as follows:
>
> **Weakness 1:** Heavy reliance on LLM outputs can lead to unpredictability and occasional hallucinations, affecting consistency in localization results. For example, different LLMs or even different LLM versions might yield different results.
>
> **Response:**
> Thank you for this valuable point. We acknowledge that LLM-based agents may suffer from unpredictability or hallucinations. However, in our system, we have incorporated several design choices to mitigate these issues and improve reliability.
>
> As illustrated in Figure 3, we introduce redundant action elimination to prevent the LLM from repeatedly generating previously executed (and potentially hallucinatory) actions. Furthermore, our agent adopts a context management framework, which enables pruning out noise and irrelevant data. Our agent has a self-consistency mechanism, where the LLM evaluates and refines its own intermediate search result to guide convergence, improving both stability and correctness. To evaluate consistency, we conducted a controlled experiment by selecting five representative instances from SWE-bench ([django__django-13551], [django__django-16255], [scikit-learn__scikit-learn-13439], [sympy__sympy-14774], [sympy__sympy-15011]). For each, we ran the agent five times and consistently observed the same function-level localization results under the same LLM configuration, demonstrating strong intra-model stability.
>
> For inter-model consistency, we agree that different models varying a lot due to their reasonability and model weight discrepancy. To solve this problem, we would focus on improving LLM consistency by exploring consistency-aware prompting, robust fine-tuning, and more powerful code index and embedding in the future.
>
> **Weakness 2:**
> The evaluation primarily focuses on SWE-bench datasets.
>
> **Response:**
> Currently, we rely on SWE-Bench, which is the state-of-the-art benchmark so far for evaluating software agents. We recently discovered additional benchmarks such as SWE-Gym and LocBench, however they were released so lately that we do not have time to review the findings before submission. In the future, we plan to expand our method to include more benchmarks and experiment with other approaches.
>
> Another reason is that the benchmark uses repo-based software, which increases the cost significantly. Experiments for 300 instances on SWE-bench lite will cost between `$500` and `$600`, not considering debugging and reproduction for different baselines. To address this issue, we will continue to work on efficient models (such as small model distillation and finetuning) to better preserve tokens.
>
> **Weakness 3:** The paper provides limited discussion on computational overhead and scalability, leaving questions about resource requirements for practical deployment.
>
> **Response:**
>
> Thank you for bringing this up. Our initial experiments primarily focused on accuracy and localization performance, so we did not perform extensive optimization for computational overhead. Currently, our system runs using API-based access to LLMs, which means that the local infrastructure requirement is minimal—a standard CPU server is sufficient to orchestrate the agent pipeline.
>
> However, we acknowledge that scalability and cost-efficiency are crucial for practical deployment. In future work, we plan to serve our own models (e.g., fine-tuned variants) using lightweight LLM-serving frameworks such as Sglang, which would likely require GPU resources (e.g., A100 or H100) depending on the model size.
>
> For additional details regarding cost analysis, including token usage and runtime cost on SWE-bench Lite, please refer to our response to Reviewer 3 (due to character limitations here). Thank you for your understanding!

---

### Decision · Program_Chairs · 2025-05-01

**Decision:**

Accept (poster)

**Comment:**

The manuscript proposes a new framework for software issue localization -- one of the long-standing challenges in software engineering. The framework is well designed and presented with rigorous empirical evaluation on some of the most challenging repository-level issue resolution benchmarks. All the reviewers call out the technical merits of the work, and I concur with the reviewers. Using ReAct style frameworks -- that are fairly general and extend to pretty much any new domain --- is the most common strategy in software agents and in particular for code issue resolution that involves navigation of large codebases, invocation of search tools, as well as reasoning over extracted code snippets. It's great to a see a well-thought out framework designed especially for the localization problem, which is a significant component to the success of coding agents.

The paper makes clear technical contributions and I recommend acceptance. There a few minor issues the reviewers raise (about in-depth analysis, ablations, cost analysis, etc.) which the authors do address in their rebuttals. I encourage the authors to revise the final version to include these analyses.